# StegoZip: Enhancing Linguistic Steganography Payload in Practice with Large Language Models

**Jun Jiang**[1,2], **Zijin Yang**[1,2], **Weiming Zhang**[1], **Nenghai Yu**[1], **Kejiang Chen**[1,2*]

1. University of Science and Technology of China, China

2. Anhui Province Key Laboratory of Digital Security, China

`{jungle0430@mail., chenkj@}ustc.edu.cn`

Codes: `https://github.com/Jungle0430/StegoZip`

## Abstract

Generative steganography has emerged as an active research area, yet its practical system is constrained by the inherent secret payload limitation caused by low entropy in generating stego texts. This payload limitation necessitates the use of lengthy stego texts or frequent transmissions, which increases the risk of suspicion by adversaries. Previous studies have mainly focused on payload enhancement through optimized entropy utilization while overlooking the crucial role of secret message processing. To address this gap, we propose *StegoZip*, a framework that leverages large language models to optimize secret message processing. StegoZip consists of two core components: semantic redundancy pruning and index-based compression coding. The former dynamically prunes the secret message to extract a low-semantic representation, whereas the latter further compresses it into compact binary codes. When integrated with state-of-the-art steganographic methods under lossless decoding, StegoZip achieves $2.5\times$ the payload of the baselines while maintaining comparable processing time in practice. This enhanced payload significantly improves covertness by mitigating the risks associated with frequent transmissions while maintaining provable content security.

## 1 Introduction

Steganography is an information-hiding technique that enables covert communication by imperceptibly modifying cover data to avoid detection by adversaries [1–3]. For example, to establish collaboration with target customers while preventing detection by competitors, a company may employ steganography to transmit commercial information through public channels by embedding trade secrets into innocuous cover data. Unlike cryptography, which secures content through encryption, steganography ensures security by eliminating physical or statistical traces of hidden information in the cover data, thereby avoiding suspicion from adversaries at the behavioral level [4–6].

Linguistic steganography, which exploits text as the most prevalent communication medium, as shown in Figure 1, typically follows two core phases during message encoding [7–9]: 1) **Message Processing:** preprocessing secret messages through compression, encryption, and format conversion. 2) **Message Embedding:** most of these methods adopt channel coding methods to embed messages while balancing imperceptibility and payload, exemplified by Syndrome-Trellis Codes (STC) [10] and Steganographic Polar Codes (SPC) [11]. After transmission of the stego texts via the public channel, authorized receivers can reconstruct the secret message by applying inverse transformations using shared keys. However, traditional modification-based methods [10, 11] invariably introduce detectable statistical discrepancies between the cover texts and the stego texts [12–15].

---

*Corresponding authors: Kejiang Chen chenkj@ustc.edu.cn;

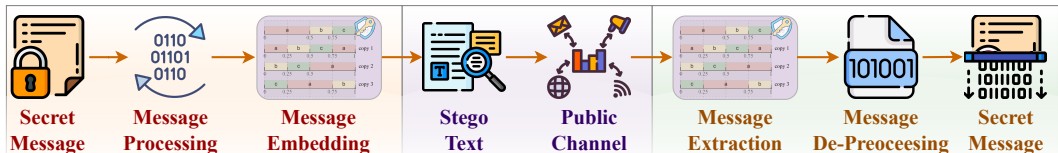

Figure 1: The general process of linguistic steganography.

With breakthroughs in generative large language models (LLMs) [16], a transformative paradigm shift has emerged in the field of steganography. These models not only generate texts that are closely aligned with human text distributions but also, more importantly, explicitly output token-level sampling probability distributions. By embedding secret messages while maintaining the original distributions and sampling randomness, it is feasible to develop provably secure steganography [6, 17–24]. For example, Discop [23] generates multiple "distribution copies" for a given probability distribution and employs the index values of these copies to encode secret messages. SparSamp [24] achieves efficient message embedding by combining sparse sampling with message-derived random numbers, significantly reducing computational complexity while maintaining security.

However, practical steganographic systems are still constrained by the relatively low secret payload, which is defined as the ratio of the secret message length to the stego text length. To transmit the secret message, the stego text often has to be considerably longer than the secret message itself. Consequently, the sender must either use lengthy stego texts or multiple transmissions to ensure the integrity of the secret message. These anomalous behaviors are more likely to expose the purpose of steganography or necessitate more complex compensatory actions.

While state-of-the-art (SOTA) methods enhance payload through iterative embedding [18, 21–24] or sampling distribution optimization [25–27], they focus mainly on the message embedding phase, i.e., how to leverage the statistical characteristics of cover texts to embed secret messages more efficiently. This singular focus, however, is still bound by the limitations of entropy and overlooks critical opportunities for message processing optimization by LLMs, particularly in terms of redundancy elimination and semantic compression of secret messages before embedding operations.

On the basis of these insights, we propose *StegoZip*, a framework designed to address the payload limitations in practical linguistic steganography through LLM-driven secret message automated processing. The framework consists of two key components: Dynamic Semantic Redundancy Pruning (DSRP) and Index-Based Compression Coding (ICC). DSRP uses LLMs to analyze semantic redundancy in original messages, dynamically removing low-information elements to generate pruned, low-semantic content. In addition, a restorer fine-tuned on public datasets can losslessly reconstruct the original semantic richness from the pruned content. Moreover, inspired by Shannon's information theory [28] and advancing existing LLM-based compression methods [29], ICC further compresses low-semantic content into compact index sequences. Furthermore, after pseudo-randomization, StegoZip maintains the security of the underlying steganographic algorithm during secret message embedding. In summary, our main contributions are as follows:

- We reveal the communication risks arising from the payload limitations in practical linguistic steganographic systems, highlighting that inefficient message processing optimization restricts the capacity of secure covert communication.

- We propose StegoZip, a framework for secret message automated processing that enhances payload in SOTA steganographic systems while maintaining security during secret message embedding.

- By incorporating StegoZip into SOTA steganographic systems, we achieve 2.5× the payload of the baselines while preserving a comparable steganographic processing time for lossless reconstruction of secret messages, demonstrating practical viability for real-world deployment.

## 2 Related Work

### 2.1 Generative Linguistic Steganography

Linguistic steganography hides secret messages within a cover text. Traditional methods, e.g., Syndrome-Trellis Codes (STC) [10], and Steganographic Polar Codes (SPC) [11], achieve this by modifying components of the cover text. However, these methods often induce statistical deviations

from the natural distribution, making the stego text vulnerable to detection by adversaries. In contrast, generative language modeling has revolutionized the field by enabling the embedding of secret messages into generative data [17]. These models learn the underlying distributions of natural language and serve as effective sampling mechanisms, producing content that is increasingly statistically indistinguishable from human text. This capability forms the foundation for secure steganography, as it allows for secret messages to be embedded during the token generation process without disrupting the statistical properties of the output [18, 30]. By iteratively sampling from explicit probability distributions over tokens, generative models ensure both security and naturalness in the resulting stego text.

Recent advances in generative linguistic steganography have capitalized on these principles. Notable examples include ADG [20], which partitions the vocabulary into clusters of similar probabilities and uses the cluster indices to represent secret messages. Meteor [21] employs range-reversible sampling that encodes messages as sampling interval offsets while compressing code length via shared prefixes. iMEC [22] attains near-theoretical embedding limits by iteratively optimizing message encoding paths based on minimum entropy coupling theory, maximizing embedding capacity with minimal distortion. Discop [23] takes a novel method by generating multiple "distribution copies" from a given probability distribution and encoding secret messages via indices of these copies. SparSamp [24] combines sparse sampling with message-derived random numbers, drastically reducing computational complexity while preserving high levels of security.

While these state-of-the-art (SOTA) methods have significantly improved payload through iterative embedding [18, 21–24] or sampling distribution optimization [25–27], they are still constrained by the inherently low entropy in the probabilities of text generation. This limitation caps payload efficiency, as the distribution of natural language tokens leaves little room for substantial expansion. However, the advent of large-scale language models presents transformative opportunities not only in optimizing the embedding stage but also in rethinking the processing and compression of secret messages. In light of this, we propose StegoZip for advanced message processing and compression.

## 3 Methodology

### 3.1 Overview

As shown in Figure 2, StegoZip operates through two LLM-driven components: Dynamic Semantic Redundancy Pruning (DSRP) and Index-Based Compressed Coding (ICC). The process begins with the use of LLMs to systematically eliminate redundant, low-information elements via semantic pruning. To prevent the removal of critical information, this step incorporates entity detection and self-checking mechanisms. Once semantic pruning is complete, the framework employs the LLM again to perform probability-driven index-based compression, producing rank sequences. These sequences undergo binary encoding and pseudo-randomization, resulting in pseudo-random bit streams. The resulting bit streams are then embedded into cover texts through a steganographic embedding algorithm, enabling secure and covert transmission over public communication channels.

The authorized receiver, equipped with prior knowledge of the steganographic embedding method, binary encoding schema, and LLM configurations, performs the reverse transformations to decode the compressed messages. After successful extraction, a shared semantic restorer $\mathcal{R}$, fine-tuned on public datasets, losslessly reconstructs the original secret messages in a context-aware manner. To maintain synchronization between the sender and receiver, the sender also uses $\mathcal{R}$ as the LLM for both DSRP and ICC. Further details of each module are described in the following sections.

### 3.2 Private Restorer in StegoZip Framework

The private restorer $\mathcal{R}$ serves as a core component of the StegoZip framework, derived through fine-tuning a base language model. In natural communication, rich semantic content enables the receiver to fully understand the sender's perspective. However, such semantic abundance introduces significant redundancy, which increases the payload burden during transmission over public channels. Thus, this redundancy limits practical applications in resource-constrained communication scenarios, e.g., covert communication. To address this, the framework harnesses the advanced language comprehension capabilities of LLMs to perform semantic pruning, retaining only the most critical elements of the secret messages for public transmission.

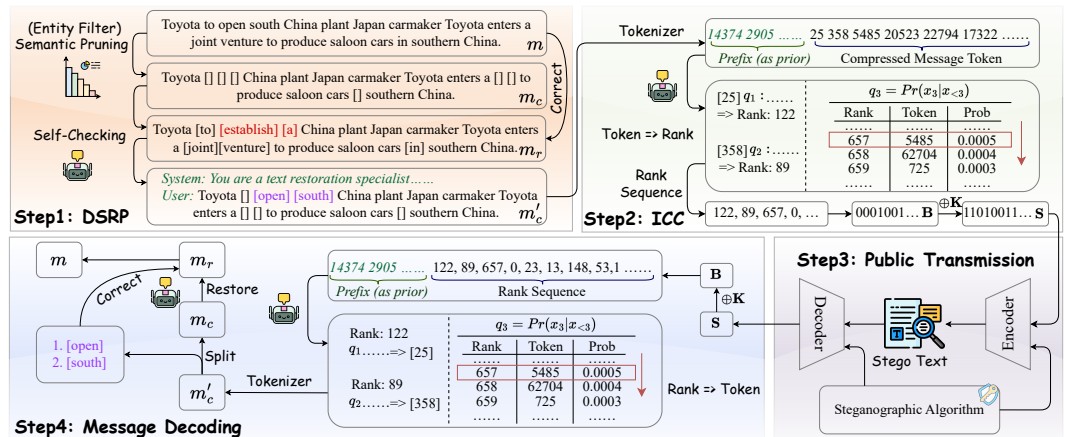

Figure 2: The framework of StegoZip comprises two core components: Dynamic Semantic Redundancy Pruning (DSRP) and Index-Based Compressed Coding (ICC).

However, relying solely on human perception to interpret the pruned messages can lead to ambiguity or misunderstanding, as the compressed content may lack sufficient contextual clarity. To address this issue, the framework harnesses the context-aware capabilities of LLMs to reconstruct the original rich semantics from the compressed messages. Notably, deploying large-scale LLMs in covert communication scenarios is often impractical because of their substantial computational and resource requirements, whereas smaller models (e.g., 3B, 7B) lack the capacity to reliably comprehend and execute restoration tasks. To balance these constraints, the framework fine-tunes a smaller base LLM, creating a private shared restorer model $\mathcal{R}$ used by both the sender and receiver to ensure lossless synchronization and significantly enhance communication fidelity. The fine-tuning process involves three key steps: Self-Information Calculation, Semantic Pruning, and Instruction Fine-tuning.

**Self-Information Calculation.** Instruction fine-tuning begins by constructing a dataset tailored for semantic restoration. This necessitates preparing paired data where the input comprises text with low semantic content, while the output retains rich semantic details. Given a public text dataset $\mathcal{D}_p$, for each sample $x_p \in \mathcal{D}_p$ designated as output, its compressed content serves as the corresponding input. This is achieved by leveraging the concept of self-information from information theory to measure the information content of each lexical unit (e.g., words or any meaningful segments derived from tokenization) in $x_p$. This metric quantifies the significance of a lexical unit based on its sampling probability as determined by the base language model. For each lexical unit $u \in x_p$ consisting of $k$ tokens $u = \{w^{(1)}, ..., w^{(k)}\}$, its self-information $I_{\text{lex}}(u)$ is calculated as:

$$I_{\text{lex}}(u) = \sum_{i=1}^{k} I\left(w^{(i)}\right) = \sum_{i=1}^{k} \left(-\log P\left(w^{(i)}\right)\right) = \sum_{i=1}^{k} \left(-\log p(w^{(i)} \mid w_{<t})\right), \quad (1)$$

where $p(w^{(i)} \mid w_{<t})$ is the conditional probability of the $t$-th token $w^{(i)}$ sampled, given its preceding tokens $w_{<t}$ as estimated by the LLM. The less likely a token is to be sampled, the greater its self-information.

**Semantic Pruning.** Once the self-information of all lexical units in $x_p$ is calculated, units with low information are removed through $\alpha$-quantile pruning:

$$\mathcal{D}_c = \left\{ x_p \odot \mathbf{1}\left(I_{\text{lex}}(u) > \tau_\alpha, \ \forall u \in x_p\right) \mid x_p \in \mathcal{D}_p \right\}, \quad (2)$$

where $\odot$ denotes element-wise multiplication, $\mathbf{1}(\cdot)$ is an indicator function, and $\tau_\alpha$ represents the $\alpha$-quantile satisfying:

$$\tau_\alpha = \inf \left\{ \tau \in \mathbb{R} \ \middle| \ \frac{1}{|x_p|} \sum_{u \in x_p} \mathbf{1}\left(I_{\text{lex}}(u) \leq \tau\right) \geq \alpha \right\}. \quad (3)$$

Figure 3: Fine-tuning restorer $\mathcal{R}$.

**Instruction Fine-tuning.** After semantic pruning, the semantic-compressed dataset $\mathcal{D}_c$, combined with the original dataset $\mathcal{D}_p$, is used to construct the instruction

fine-tuning dataset $\mathcal{D}_{ft}$ through template-based pairing, as illustrated in Figure 3. The base LLM is fine-tuned to reconstruct rich semantics by inserting words within square brackets "[ ]", without altering or deleting any existing content in the compressed text. This ensures precise restoration while preserving the original structure. The fine-tuned restorer $\mathcal{R}$, shared by both sender and receiver, enables lossless and synchronized semantic reconstruction in the StegoZip framework.

### 3.3 Dynamic Semantic Redundancy Pruning

With the shared private restorer $\mathcal{R}$ established, the StegoZip module integrates seamlessly into the steganographic system, employing a dynamic pruning mechanism through two phases similar to the restorer fine-tuning process: Self-Information Calculation and Semantic Pruning.

For a secret message $m$, its compressed form $m_c$ is derived using $\mathcal{R}$ and self-information processing: $m_c = m \odot \mathbf{1}\left(I_{\text{lex}}(u) > \tau'_\alpha\right)$. Here, $\tau'_\alpha$ is a dynamic threshold that adjusts based on the average self-information of $m$ and the empirical fine-tuning dataset $\mathcal{D}_p$:

$$\tau'_\alpha = \tau_\alpha \cdot \left(1 - \eta \cdot \frac{\bar{I}(m) - \bar{I}(\mathcal{D}_p)}{\bar{I}(\mathcal{D}_p)}\right), \tag{4}$$

where $\tau_\alpha$ is the predefined threshold from Eq. (3). $\bar{I}(m)$ is the average self-information of secret message $m$ and $\bar{I}(\mathcal{D}_p)$ is the average self-information of $\bar{I}(x_p)$ from the public dataset $\mathcal{D}_p$. This ensures dynamic adjustment of the pruning ratio based on the entropy of $m$, preventing excessive pruning in short, high-entropy texts. Extremely high self-information values ($\infty$) are ignored.

However, this process risks losing critical information, e.g., names, numbers, or other key entities. To address this, entity detection is performed prior to pruning, ensuring that essential details are preserved intact, with their self-information designated as infinity $\infty$. Additionally, to guarantee that the recipient can reconstruct the message without loss, the sender conducts a self-checking process. Using the restorer $\mathcal{R}$, the sender attempts to reconstruct the original secret messages. Any portions that cannot be accurately aligned are replaced with their original content and marked with square brackets "[ ]", as shown in Figure 2. Through this careful preprocessing, the resulting compressed secret messages $m'_c$ retain low semantic redundancy while safeguarding crucial elements. Moreover, having been exposed to numerous instances of compressed text during fine-tuning, $\mathcal{R}$ is better equipped to compress text when using the same prefix in subsequent encoding tasks through ICC.

### 3.4 Index-Based Compressed Coding

After dynamic semantic pruning, the compressed message $m'_c$ is transformed into binary codes via probability-driven index encoding. Inspired by LLMzip [29], this method leverages the token prediction capabilities of the LLM to achieve high compression ratios.

Let the tokenized sentence be represented as $\mathbf{W}_c = \{w_1, ..., w_s\}$ of length $s$, where each $w_t \in \mathbf{W}_c$ denotes as a token. The rank of each token $w_t$ is defined as its sorted index when the vocabulary $\mathcal{V}$ is ordered by descending conditional probability:

$$r(w_t) = 1 + \sum_{w' \in \mathcal{V} \setminus \{w_t\}} \mathbf{1}\left(p\left(w' \mid w_{<t}\right) > p\left(w_t \mid w_{<t}\right)\right), \tag{5}$$

where $\mathbf{1}(\cdot)$ is an indicator function. This probability-driven token-rank mapping ensures efficient compression, as tokens with higher probabilities are assigned lower ranks. Since the fine-tuned restorer $\mathcal{R}$ has been exposed to numerous instances of semantic pruning text with the same template prefix, it is particularly effective at encoding such texts, thereby achieving a higher compression rate.

The rank sequence is then converted into bit format $\mathbf{B}$ via Huffman encoding, a lossless compression method that minimizes the bit form of frequent ranks. To ensure secure steganographic embedding, the bit sequence $\mathbf{B}$ is pseudo-randomized. This is achieved by performing an XOR operation between $\mathbf{B}$ and a pseudo-random binary key $\mathbf{K}$, which is generated by a secure stream encryption algorithm, e.g., ChaCha20 [31]. The resulting pseudo-random bit stream $\mathbf{S}$ is then embedded into the cover text via a secure steganographic embedding function, ensuring provable security while maintaining the integrity of the compressed message. For a detailed proof of security, please refer to Appendix A.

### 3.5 Secret Message Restoration

The decoding framework enables lossless reconstruction of the secret message through invertible transformations mirroring the encoding process. For the stego text $x_s$, the embedded bit stream $\mathbf{S}$ is extracted via the negotiated steganographic extraction function. The receiver then reverses the pseudo-randomization using cryptographically synchronized parameters and Huffman decoding with the same shared codebook to obtain the original rank sequences. Furthermore, the receiver replicates the index-based compression coding generation process via inverse rank-token mapping, converting each rank into its corresponding token to reconstruct the semantic pruning message $m_c'$.

Since $m_c'$ may lack sufficient semantic detail for complete comprehension, the shared restorer $\mathcal{R}$ employs instruction-guided semantic expansion to restore the full secret message. During this step, $m_c'$ is split into the compressed representation $m_c$ and the portion marked during the sender's self-checking process in the DSRP phase. The restorer synchronizes with the sender's self-checking process, producing the intermediary text $m_r$ with the same errors. Finally, the marked portions are used to correct any errors in $m_r$, ensuring the complete restoration of the original secret message.

## 4 Experiments

### 4.1 Implementation Details

**LLMs.** In this paper, we utilize two widely used LLMs: Qwen2.5-7B [32] and DeepSeek-R1-Distill-Llama-8B (DS-Llama-8B) [33] for index-based coding and restoration tasks, and greedy sampling is employed. To prevent fine-tuning that could compromise the security of the original generative steganography algorithm, we use separate LLaMA2-7B [34] for stego text generation. In this process, random sampling with a temperature of 0.9 is applied, without incorporating top-$p$ or top-$k$ sampling. We use LoRA [35] to fine-tune the base LLMs for two epochs, and more detailed experimental settings are shown in the Appendix C.

**Datasets.** Text datasets are used to fine-tune the restorer and generate stego text. For fine-tuning, the IMDb [36] dataset (average length: 1,300 characters) is split into 25,000 training texts and 25,000 test texts, but only 2,000 are randomly sampled for testing. For the AGNews [37] dataset (average length: 241 characters), only the "business" category is selected and divided into 30,000 training texts and 1,900 test texts. For stego text generation, the WikiText-2-v1 [38] dataset is used. Texts are randomly sampled, and the first two sentences are extracted as prompts to guide generation.

**Baselines.** In the main experiment, the parameters for the proposed Dynamic Semantic Redundancy Pruning are set to $\alpha = 0.4$ and $\eta = 1.0$. To the best of our knowledge, existing linguistic steganographic systems do not specifically address message processing. Therefore, we adopt a common setup, using Huffman compression after tokenizing the original secret messages as the baseline message processing. The tokenizers [32, 33] we use employ Byte Pair Encoding (BPE) [39] based on bytes, which inherently provides some degree of compression. We consider SOTA methods for the underlying generative steganography: Discop [23] and SparSamp [24]. For a fair comparison, we evaluate performance using sentences that do not have token ambiguity [40, 41].

**Evaluation metrics.** We evaluate StegoZip from both efficiency and semantic retention perspectives:

**1. Efficiency:** We divide the efficiency into payload and processing time. The payload refers to the ratio of the secret message length to the stego text length, which is the most important metric for assessing StegoZip's compression capability. The processing time encompasses the average encoding time, which spans from processing the secret message to the completion of generating the stego text, and the average decoding time, which involves extracting the bit stream from the stego text and restoring the original secret message.

**2. Semantic Retention:** We evaluate the semantic retention from restored and compressed messages at the word and semantic levels via the metric $\text{Rouge}_1$ [42] and BERTScore [43]. $\text{Rouge}-1$ calculates the proportion of single words from the original secret message that appear in the target secret message, whereas BERTScore, which leverages BERT's contextual embeddings to calculate the similarity between the original and target secret messages, provides a measure of semantic similarity. Higher values of these metrics are better.

Table 1: The efficiency of StegoZip: StegoZip significantly enhances the payload while maintaining a comparable steganographic processing time cycle.

| Model | Dataset | Steganography | Payload (%) ↑ | Encoding Time ($s$) ↓ | Decoding Time ($s$) ↓ |
|---|---|---|---|---|---|
| Qwen2.5-7B [32] | AGNews [37] | Discop [23] + StegoZip | 18.91 45.23 (↑ 26.32) | 14.82 12.94 (↓ 1.88) | 14.93 12.73 (↓ 2.20) |
| | | SparSamp [24] + StegoZip | 19.03 44.49 (↑ 25.46) | 10.55 11.23 (↑ 0.68) | 10.61 11.15 (↑ 0.54) |
| | IMDb [36] | Discop [23] + StegoZip | 18.30 49.64 (↑ 31.34) | 70.20 56.62 (↓ 13.58) | 69.21 54.91 (↓ 14.3) |
| | | SparSamp [24] + StegoZip | 19.12 49.74 (↑ 30.62) | 63.12 55.67 (↓ 7.45) | 63.04 52.36 (↓ 10.68) |
| DS-Llama-8B [33] | AGNews [37] | Discop [23] + StegoZip | 18.75 44.83 (↑ 26.08) | 14.98 12.33 (↓ 2.65) | 14.30 11.95 (↓ 2.35) |
| | | SparSamp [24] + StegoZip | 18.48 43.60 (↑ 25.12) | 10.16 10.62 (↑ 0.46) | 10.18 10.61 (↑ 0.43) |
| | IMDb [36] | Discop [23] + StegoZip | 18.33 49.31 (↑ 30.98) | 68.15 55.91 (↓ 12.24) | 69.23 53.47 (↓ 15.76) |
| | | SparSamp [24] + StegoZip | 19.76 49.41 (↑ 29.65) | 61.55 49.56 (↓ 11.99) | 60.93 46.46 (↓ 14.47) |

Table 2: Before StegoZip can reconstruct secret messages losslessly (achieving a metric score of 1.0000), compressed secret messages still retain a high degree of semantic preservation. Below, R / C denotes the semantic retention of restored/compressed secret messages relative to the original.

| Model | Dataset | $Rouge_1$ (R / C) | BERTScore (R / C) |
|---|---|---|---|
| Qwen2.5-7B [32] | AGNews [37] | **1.0000**/0.8485 | **1.0000**/0.9633 |
| | IMDb [36] | **1.0000**/0.9445 | **1.0000**/0.9835 |
| DS-Llama-8B [33] | AGNews [37] | **1.0000**/0.8658 | **1.0000**/0.9664 |
| | IMDb [36] | **1.0000**/0.9185 | **1.0000**/0.9758 |

## 4.2 Main Performance of StegoZip

**Efficiency of StegoZip.** The experimental results in Table 1 demonstrate that the proposed StegoZip framework significantly enhances the original steganographic system, **achieving a 2.5× the payload of the baselines.** This improvement is attributed to the integration of Dynamic Semantic Redundancy Pruning (DSRP) and probability-driven Index Compression Coding (ICC), which collaboratively compresses lexical units in secret messages with high efficiency. When dealing with longer secret messages (the average character count of the IMDb dataset is five times that of the AGNews dataset), StegoZip demonstrates better compression performance due to the availability of richer contextual information. Complementing these results, Table 2 provides a rigorous evaluation of message fidelity, where both the restored and compressed secret messages are compared to the original using $Rouge_1$ (word-level similarity) and BERTScore (semantic-level alignment). The results confirm that StegoZip achieves lossless reconstruction of secret messages at the receiver end, whereas the compression process preserves the core semantic meaning of the original content. Overall, the StegoZip framework harnesses the advanced semantic understanding capabilities of LLMs to eliminate low-information units in secret messages, encoding them into a densely packed and secure binary bit stream. This ensures efficient representation without compromising the lossless decoding process. By doing so, the method significantly reduces the payload burden for covert communication over public channels, while simultaneously enhancing both security and efficiency.

**Time Consumption.** Although StegoZip introduces additional preprocessing steps, including Self-Checking, ICC, De-ICC, and Restore, which may appear time-intensive, the results in Table 1 and Figure 4 reveal that the optimized payload efficiency ensures a **comparable steganographic embedding and extraction time**. This is attributed to the reduced amount of binary code that needs to be embedded per stego text, significantly decreasing the time required for both the embedding and

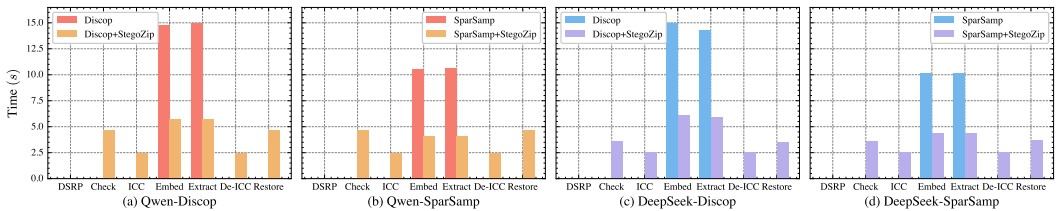

Figure 4: Comparison of time consumption across different stages of steganographic processing.

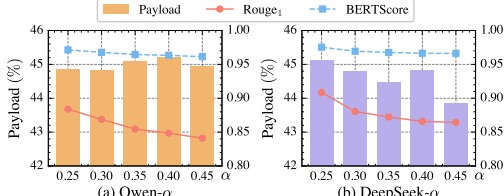

Figure 5: Effect of pruning quantile $\alpha$.

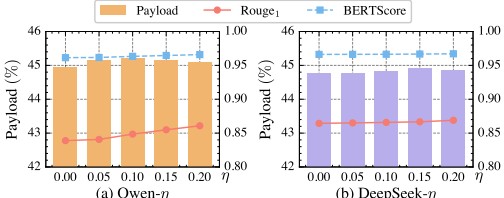

Figure 6: Effect of adaptive coefficient $\eta$.

Table 3: Efficiency comparison of LLMs of varying sizes in compression and restoration.

| Model Size | Payload (%)↑ | Encoding Time (s)↓ | Decoding Time (s)↓ | Rouge$_1$ ↓ | BERTScore ↓ |
|---|---|---|---|---|---|
| Qwen2.5-0.5B | 40.21 | **8.53** | **8.56** | 0.9251 | 0.9801 |
| Qwen2.5-1.5B | 41.15 | 9.23 | 9.08 | 0.8982 | 0.9753 |
| Qwen2.5-3B | 41.92 | 9.56 | 9.47 | 0.8884 | 0.9705 |
| Qwen2.5-7B | 45.23 | 12.94 | 12.73 | 0.8485 | 0.9633 |
| Qwen2.5-14B | **46.09** | 21.54 | 21.51 | **0.8428** | **0.9611** |

extraction processes. Moreover, the increased payload efficiency enhances security by reducing the amount of stego text required for transmission. At the behavioral level, this reduction minimizes the likelihood of arousing suspicion from adversaries under equivalent communication requirements when compared with traditional methods. These advancements position StegoZip as a practical and effective solution for balancing capacity, efficiency, and security in linguistic steganography systems.

## 4.3 Parameter Tuning and Resultant Impact

**Pruning Quantile $\alpha$.** We investigate the impact of the self-information pruning quantile $\alpha$ on both the steganographic payload and the semantic preservation of compressed secret messages. Within the range $\alpha \in [0.25, 0.45]$, as shown in Figure 5, the key semantic information in the compressed text is progressively reduced as $\alpha$ increases. A moderate pruning ratio enhances the payload as the pruning effect becomes more pronounced. However, excessive pruning disrupts the restorer $\mathcal{R}$'s ability to effectively reconstruct the original semantics. This leads to increased errors during self-checking and more extensive corrections, ultimately resulting in a decreased payload. Striking an optimal balance for $\alpha$ is therefore critical, as it directly determines the trade-off between payload and semantic fidelity.

**Adaptive Coefficient $\eta$.** The adaptive coefficient $\eta$ plays an important role in shaping the performance of StegoZip within the DSRP module. As shown in Figure 6, the model navigates the trade-off between compression efficiency and semantic resilience as $\eta$ increases. Lower values of $\eta$ favor aggressive compression, maximizing efficiency but risking over-compression, which may result in the loss of essential semantic information and hinder accurate restoration. Conversely, higher $\eta$ values are more cautious, prioritizing the preservation of semantic integrity at the cost of reduced compression efficiency. Moreover, different samples exhibit varying levels of complexity and information density, necessitating dynamic adjustments to $\eta$ to meet diverse compression demands. This adaptability ensures that StegoZip achieves a balance between compression and semantic restoration, enhancing its versatility across diverse scenarios.

**Model Size of LLMs.** We examine the influence of large language model size on performance in compression and restoration tasks, and the results are shown in Table 3. As model size increases, the payload clearly tends to increase. This can be attributed to larger models, with their more complex architectures and greater number of parameters, being better equipped to capture richer semantic

Table 4: The generalization of the Restorer $\mathcal{R}$ across different domain datasets.

| TrainSet\TestSet | AGNews-business | AGNews-world | AGNews-Mixed | IMDb | Mixed |
|---|---|---|---|---|---|
| AGNews-business | 45.23 | 45.32 | 43.48 | 50.32 | 47.26 |
| AGNews-world | 44.90 | **47.69** | 44.28 | 49.53 | 46.94 |
| AGNews-Mixed | **46.01** | 47.03 | **45.08** | **50.49** | **47.70** |
| IMDb | 40.85 | 41.31 | 41.59 | 49.74 | 44.87 |
| Mixed | 44.82 | 45.07 | 43.87 | 50.01 | 45.73 |

Table 5: Ablation study on the components of StegoZip, where "w/o" indicates not employed.

| Model | Discop+StegoZip | w/o DSRP | w/o instruction in ICC | w/o $\mathcal{R}$ in ICC | w/o ICC | Discop |
|---|---|---|---|---|---|---|
| Qwen2.5-7B | **45.23** | 31.78 | 42.71 | 40.05 | 23.17 | 18.19 |
| DS-Llama-8B | **44.83** | 30.22 | 41.13 | 38.91 | 22.54 | 18.30 |

information and patterns. As a result, they can effectively prune more redundant semantics during compression. However, the encoding and decoding times also increase with model size. Hence, in practical applications, larger models with higher payloads are better suited for tasks where real-time processing is not critical. On the other hand, smaller models, which offer faster processing times, are more appropriate for scenarios requiring quick responses.

## 4.4 Cross-Domain Transferability Analysis

Initially, fine-tuning the small-scale LLMs aimed solely to enable them to understand the restoration task, without accounting for the domain of the secret messages. Nonetheless, given the specialized nature of covert communications, interacting parties can often predict the domain of the covert information. We analyze the impact of domain differences between the dataset used to train the restorer and the test set on the payload, as shown in Table 4. For this study, we compare categories including *business* and *world* within the AGNews dataset [37], which share some similarities, against the IMDb dataset [36], which differs significantly in both content and style. Additionally, we test a mixed configuration where both datasets are combined in equal proportions. While cross-domain differences do not affect the lossless reconstruction of messages, they can slightly reduce the payload, and hybrid training substantially mitigates this effect. These results highlight the resilience of the StegoZip across domains, with hybrid training maintaining a high payload despite domain shifts.

## 4.5 Ablation Study

We conduct ablation experiments on the AGNews dataset [37] to assess the importance of each component in the StegoZip, as shown in Table 5. The results confirm that both DSRP and ICC are critical to the framework's performance. Notably, incorporating an instruction template during ICC enables the restorer $\mathcal{R}$ to predict subsequent compressed content more accurately. This improvement can be attributed to providing the restorer with a priori knowledge of the format and content of compressed secret messages, particularly tokens used for marking. These findings underscore the significance of each component in achieving both high compression and restoration fidelity.

## 4.6 Steganalysis Experiment

To further verify the security of StegoZip, we conduct steganalysis experiments. We generate 1,000 cover texts and 1,000 stego texts using LLaMA2-7B [34] and employ three steganalysis analyzers: TS-FCN [44], LSTMATT [45], and SeSy [12]. The datasets are split into training, validation, and testing sets with a ratio of 3:1:1. The experiments are configured with a learning rate of $3 \times 10^{-5}$ and trained for five epochs. To ensure robustness, the process is repeated five times, and the average accuracy on the test set is used as the steganalysis accuracy metric. For the restorer $\mathcal{R}$, we use the fine-tuned Qwen2.5-7B [32] model. It is important to note that the provable security here refers to being indistinguishable from normal generated text, not from human-written text. Provably secure steganography involves disguising steganographic behavior as normal generated text, as detailed in Appendix A and in previous works [23, 24].

Table 6: Steganalysis accuracy with and without using StegoZip.

| Method | TS-FCN [44](%) | LSTMATT [45](%) | SeSy [12](%) |
|---|---|---|---|
| Discop | 51.25±2.77 | 50.70±3.02 | 49.20±2.23 |
| Discop + StegoZip | 50.90±2.15 | 52.35±2.71 | 51.30±0.92 |
| SparSamp | 51.40±1.64 | 49.85±1.99 | 49.10±2.08 |
| SparSamp + StegoZip | 49.80±1.42 | 52.35±1.57 | 50.05±1.66 |

The classification accuracy of steganalysis is close to 50%, as shown in Table 6, indicating that stego texts and cover texts cannot be reliably distinguished. The experimental results demonstrate that StegoZip maintains a comparable security level to scenarios without using StegoZip. This further validates the consistency of the security of steganographic systems equipped with StegoZip with that of existing underlying steganographic algorithms.

## 5   Limitations

While StegoZip significantly enhances the payload of linguistic steganographic systems, its reliance on LLMs for message compression and reconstruction inevitably introduces additional computational overhead, even if the overall steganographic processing time remains comparable. Furthermore, its dependency on high-precision LLMs limits its applicability in resource-constrained environments or scenarios lacking advanced computational infrastructure [46]. Future research could aim to optimize StegoZip's computational efficiency, making it more adaptable to diverse operational contexts.

## 6   Conclusion

In this paper, we introduce StegoZip, a novel framework that leverages large language models for dynamic semantic redundancy pruning and index-based compression coding. By incorporating it into advanced steganographic systems, we achieve a payload that is 2.5 times the original size while maintaining comparable steganographic processing time. This advancement not only enhances the efficiency of the steganographic embedding process but also reduces the frequency of communication between parties, significantly lowering the risk of suspicion by adversaries. StegoZip paves the way for secure and efficient covert communication, demonstrating the transformative potential of advanced processing techniques in modern steganographic frameworks.

## 7   Acknowledgements

This work was supported in part by the National Natural Science Foundation of China under Grant 62472398, Grant U2336206, Grant U2436601, and Grant 62402469.

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

## A  Proof of the Security of StegoZip

### A.1  Provably Secure Steganography

Empirical steganographic schemes (e.g., STC [10] and SPC [11]) inherently permit an adversary to distinguish between cover and stego texts with a non-negligible advantage, undermining their security. In contrast, provably secure steganography aims to either eliminate this advantage completely, achieving information-theoretic security [5], or reduce it to a negligible level within the framework of computational security [6].

Let the cover channel, denoted as $C_{\mathcal{H}}$, represent the conditional probability distribution of cover signals $C$ given the history $\mathcal{H}$. Assuming access to a perfect sampler $M$ that precisely adheres to the distribution $C_{\mathcal{H}}$, we define $M_b^{C_{\mathcal{H}}}$ as the process that samples a segment of the cover output of length $b$. A steganographic system (stegosystem) is formally described as a triple of algorithms $(\mathsf{KGen}, \mathsf{Embed}, \mathsf{Extract})$, corresponding to key generation, embedding, and extraction, respectively. The key generation algorithm produces a key $\mathcal{K}$ as follows:

$$\mathcal{K} \leftarrow \mathsf{KGen}(\kappa), \tag{6}$$

where $\kappa$ is a security parameter. Given a secret message $m$, a history $\mathcal{H}$, and access to the sampler $M$, the embedding algorithm generates a sequence:

$$s_1 \,|\, s_2 \,|\, \cdots \,|\, s_s \leftarrow \mathsf{Embed}^M(\mathcal{K}, m, \mathcal{H}), \tag{7}$$

where the sequence length is $s$. The extraction algorithm, which uses the negoriated key $\mathcal{K}$ and history $\mathcal{H}$, reconstructs the hidden message $\widetilde{m}$ from the stego text:

$$\widetilde{m} \leftarrow \mathsf{Extract}^M\Big(\mathcal{K}, \ \mathsf{Embed}^M(\mathcal{K}, m, \mathcal{H}), \ \mathcal{H}\Big). \tag{8}$$

Within computational security, the notion of a distinguishing game is employed, wherein an adversary $W$ attempts to differentiate the cover distribution $C$ from the stego distribution $S$. Specifically, the adversary seeks to distinguish between (i) samples generated through secret-message-driven embedding Embed and (ii) samples generated by a standard random sampling process $O$ consistent with the cover distribution. The adversary's distinguishing advantage is defined as:

$$\mathcal{A}_{C,S}^{\mathsf{SS}}(W) = \left| \Pr_{\mathcal{K}, M, \mathsf{Embed}} \left[ W^{M, \mathsf{Embed}(\mathcal{K}, \cdot, \cdot)} = 1 \right] - \Pr_{M, O} \left[ W^{M, O(\cdot, \cdot)} = 1 \right] \right|, \quad (9)$$

where the probabilities are computed over the randomness of $\mathcal{K}$, $M$, Embed, and $O$. A stegosystem is deemed computationally secure if, for every probabilistic polynomial-time adversary $W$, this advantage remains negligible in the security parameter $\kappa$:

$$\mathcal{A}_{C,S}^{\mathsf{SS}}(W) < \mathsf{negl}(\kappa). \quad (10)$$

## A.2 StegoZip Maintains the Security of the Underlying Steganographic Algorithm

To ensure that the bitstream processed by StegoZip can be securely embedded using established provably secure steganographic embedding methods, the bitstream must first undergo pseudo-randomization. This is typically achieved through an XOR operation with a pseudo-random binary keystream generated by a secure stream encryption algorithm, e.g., CHACHA20 [31]. Below, we provide a proof that this process maintains the security of the underlying steganographic algorithm.

The proof proceeds by contradiction. Assume that the stego text $x_s$ generated by the StegoZip framework and the normally generated cover text $x_c$ are distinguishable, that is:

$$\left| \Pr\left[ \mathcal{A}_{\mathcal{D}}(x_s) = 1 \right] - \Pr\left[ \mathcal{A}_{\mathcal{D}}(x_c) = 1 \right] \right| \geq \delta, \quad (11)$$

where $\delta$ is non-negligible with respect to the key $\mathcal{K}$. For tokens $w$ generated at each step under history $\mathcal{H}$, this assumption implies the following:

$$\left| \Pr\left[ \mathcal{A}_{P_{\mathcal{H}}}(w_s) = 1 \right] - \Pr\left[ \mathcal{A}_{P_{\mathcal{H}}}(w_c) = 1 \right] \right| \geq \delta. \quad (12)$$

In a generative provably secure steganographic embedding algorithm, the addition of a token to the stego sequence involves two sampling stages. The first stage, which is controlled by the steganographic embedding algorithm and driven by message bits, determines the region of sampling. For example, ADG [20] selects a cluster, Discop [23] determines which distribution copies to access, and SparSamp [24] decides whether to apply an offset. The second stage occurs within $v_{stega}$, driven by pseudo-random numbers $r$ generated by a cryptographically secure pseudo-random number generator (CSPRNG). For simplicity, we let $\mathcal{S}(r, P)$ denote sampling from $P$ via $r$, and let $\mathcal{E}(m, P)$ represent the steganographic embedding algorithm, where $m$ is the secret message encrypted via the cryptographic algorithm mentioned before.

Using the law of total probability, the adversary's decision on $w_s$ can be expressed as:

$$\begin{aligned} \Pr\left[ \mathcal{A}_{P_{\mathcal{H}}}(w_s) = 1 \right] &= \Pr\left[ \mathcal{A}_{P_{\mathcal{H}}}(w_s) = 1 \mid \mathcal{A}_{P_{\mathcal{H}}}(v_{stega}^{w_s}) = 1 \right] \Pr\left[ \mathcal{A}_{P_{\mathcal{H}}}(v_{stega}^{w_s}) = 1 \right] \\ &\quad + \Pr\left[ \mathcal{A}_{P_{\mathcal{H}}}(w_s) = 1 \mid \mathcal{A}_{P_{\mathcal{H}}}(v_{stega}^{w_s}) = 0 \right] \Pr\left[ \mathcal{A}_{P_{\mathcal{H}}}(v_{stega}^{w_s}) = 0 \right], \end{aligned} \quad (13)$$

where $w_s = \mathcal{S}(r, p_{stega}^m)$ and $v_{stega}^{w_s} = \mathcal{E}(m, P)$. Similarly, since the probabilities of tokens remain unchanged before and after steganographic embedding, a single normal generation of samples is equivalent to two independent sampling processes using random numbers. For the normal cover text, the adversary's decision on $w_c$ can be expressed as:

$$\begin{aligned} \Pr\left[ \mathcal{A}_{P_{\mathcal{H}}}(w_c) = 1 \right] &= \Pr\left[ \mathcal{A}_{P_{\mathcal{H}}}(w_c) = 1 \mid \mathcal{A}_{P_{\mathcal{H}}}(v_{cover}^{w_c}) = 1 \right] \Pr\left[ \mathcal{A}_{P_{\mathcal{H}}}(v_{cover}^{w_c}) = 1 \right] \\ &\quad + \Pr\left[ \mathcal{A}_{P_{\mathcal{H}}}(w_c) = 1 \mid \mathcal{A}_{P_{\mathcal{H}}}(v_{cover}^{w_c}) = 0 \right] \Pr\left[ \mathcal{A}_{P_{\mathcal{H}}}(v_{cover}^{w_c}) = 0 \right], \end{aligned} \quad (14)$$

where $v_{cover}^{w_c} = \mathcal{S}(r_1, P)$ and $w_c = \mathcal{S}(r_2, p_{cover})$. Here, $r_1$ and $r_2$ are independent random variables corresponding to the two-stage sampling process, each of which is uniformly distributed as $r_1, r_2 \overset{\text{i.i.d.}}{\sim} U[0, 1)$.

To satisfy Eq. 12, the adversary must be able to distinguish between $\mathcal{E}(m, P)$ and $\mathcal{S}(r_1, P)$ in polynomial time, or between $\mathcal{S}(r, p_{stega}^m)$ and $\mathcal{S}(r_2, p_{cover})$. However, since a CSPRNG is used in our synchronous sampling function, the generated $r$ is indistinguishable from a truly random sequence in polynomial time. Thus, the latter condition cannot hold.

Regarding the former condition, the bitstream $m$ generated by StegoZip is pseudo-randomized, rendering it indistinguishable from a uniform random bitstream within polynomial time. Moreover, the employed provably secure steganographic algorithm guarantees that the adversary cannot achieve a non-negligible advantage $\delta'$ in distinguishing between $\mathcal{E}(m, P)$ and $\mathcal{S}(r_1, P)$ in polynomial time:

$$\left| \Pr\left[ \mathcal{A}_{P_{\mathcal{H}}}(v_{stega}^{w_s}) = 1 \right] - \Pr\left[ \mathcal{A}_{P_{\mathcal{H}}}(v_{cover}^{w_c}) = 1 \right] \right| < \mathsf{negl}(\kappa). \tag{15}$$

Since neither condition can be satisfied, Eq. 12 is invalid, contradicting the assumption in Eq. 11. Therefore, StegoZip, when integrated with any computationally secure steganographic algorithm, constitutes a computationally secure steganographic system. □

# B Ethics Considerations

In this paper, we propose the StegoZip framework to enhance the payload of linguistic steganography, specifically for scientific research and educational purposes. We strictly adhere to established scientific research regulations to ensure data privacy and security throughout the experimental process, and we rigorously avoid any violation of personal privacy or engagement in illegal activities. We are committed to responsibly advancing academic research in information security and ensuring that our contributions positively impact society.

# C More Experiment Settings

## C.1 Model Fine-tuning

The restorer is fine-tuned to help small-scale LLMs grasp the restoration task effectively. To maintain consistency across multiple runs, a random seed of 42 is employed. The training process adopts a micro-batch size of 32 and an overall batch size of 64, necessitating the use of gradient accumulation steps computed as the ratio of the overall batch size to the micro-batch size. The model is trained for 2 epochs, with the learning rate set at $3 \times 10^{-5}$.

For the IMDb dataset, the sequence length is cut or filled at 512 tokens, while for the AGNews dataset, the sequence length is limited to 256 tokens. The LoRA (Low-Rank Adaptation) parameters are carefully configured with LORA_R = 16, LORA_ALPHA = 32, and a dropout rate of 0.05. Fine-tuning targets specific modules, including $\{q\_proj, k\_proj, v\_proj, o\_proj, gate\_proj, up\_proj, down\_proj\}$. To optimize computational efficiency while maintaining model performance, the model is loaded in int8 precision. The dataset is split into training and validation sets in a ratio of 4:1.

The instruction used for fine-tuning, illustrated in Figure 3, is crafted to guide the model in performing text restoration tasks. Specifically, the instruction reads: *"You are a text restoration specialist. Your task is to ONLY fill in the missing content within square brackets [ ] in the input text. Requirements: 1. Strictly preserve all existing text and punctuation outside brackets. 2. Maintain original text structure and formatting."* To ensure stable fine-tuning and prevent the model from altering content outside the "[ ]", 10% of the fine-tuning dataset is constructed with inputs and outputs that remain identical.

To enable the model to reliably identify the conclusion of its response, an "[END]" marker is appended to the end of responses in the training dataset. During inference, this "[END]" flag also serves as a termination signal for the model's outputs.

All our experiments are conducted on a hardware platform equipped with an Intel(R) Xeon(R) Gold 6130 CPU operating at 2.10 GHz, 256 GB of RAM, and NVIDIA A6000 GPU cards. Training the Qwen2.5-7B [32] and DeepSeek-R1-Distill-Llama-8B [33] models on the AGNews dataset requires approximately 6 hours, while training on the IMDb dataset takes around 10 hours.

## C.2 Model Inference

During the model inference phase, the model is loaded in the FP32 format to accommodate the high-precision probabilistic sorting required by index compression coding. When the restorer is tasked with reconstructing high semantic information, greedy sampling is employed to ensure deterministic and accurate outputs. We do not use the chain-of-thought feature of the DeepSeek-R1-Distill-Llama-8B [33] model, given its status as a widely adopted text-generation model. Its language patterns and knowledge acquired during pre-training remain highly effective even after fine-tuning, particularly for restoration tasks requiring the handling of complex semantics. Furthermore, if inference functions or other advanced features are to be introduced in the future, the model's existing architectural foundation provides an extensible and optimizable framework, facilitating seamless integration and enhancement. After fine-tuning, the model will not initiate its response with "<think>\n" at the beginning of every output.

## C.3 Marker Compression Processing

To manage vacant positions awaiting restoration, we represent them using brackets "[ ]". However, during the Index-Based Compressed Coding process, instead of transforming the "[ ]" into its rank, we replace it with a separator symbol to generate a rank sequence optimized for better compression.

For example, consider the sequence "$(x_1, x_2, [\,], x_4, x_5, [\,], [\,], [\,], x_9)$", where $x_i$ represents a lexical unit. Initially, the positions of "[ ]" are recorded, and the sequence is transformed into a rank sequence "$39 \sqcup 23 \sqcup 3 \sqcup 19 \sqcup 6 \sqcup 4 \sqcup 5 \sqcup 3 \sqcup 7$", which is stored as a string with ranks separated by the symbol "$\sqcup$". Next, the recorded positions of "[ ]" are replaced with consecutive separators, resulting in a modified sequence "$39 \sqcup 23 \sqcup \sqcup 19 \sqcup 6 \sqcup \sqcup \sqcup \sqcup \sqcup 7$". The vacancies are then identified by the successive separators.

This processed sequence is subsequently subjected to Huffman encoding, yielding a highly compressed bit stream. By adopting this method, we ensure accurate localization of vacant positions while significantly improving the compression rate.

## C.4 Steganography

**Discop.** We adhere strictly to the official open-source repository of Discop[2]. Specifically, we utilize the Huffman tree-enhanced version to maximize the payload and seamlessly integrate it into our codebase.

**SparSamp.** Similarly, we follow the official open-source repository of SparSamp[3], adopting the configuration with *block_size* $= 32$ and incorporating it into our codebase.

**Stego Texts Generation.** During the generation of stego text, random sampling with a temperature of 0.9 is applied to introduce controlled variability, while avoiding the use of top-$p$ or top-$k$ sampling techniques to maintain alignment with the original framework.

## C.5 Token Ambuguity

Mainstream provably secure generative linguistic steganographic methods [20–24] rely on the precise alignment of token paths between the sender and receiver to achieve successful decoding. However, this requirement becomes challenging in LLMs because of token ambiguity caused by byte-pair encoding (BPE) [39]. BPE generates non-prefix-free vocabularies, allowing multiple tokenization paths to represent the same input sequence. To address this issue, alternative strategies such as word-based tokenizers, character-based tokenization, or disambiguation algorithms [40, 41] can be employed to reduce token ambiguity. In our main experiments, we ensure fairness by evaluating performance via sentences without token ambiguity.

---

[2]The repository of Discop [23] is available at: `https://github.com/comydream/Discop`
[3]The repository of SparSamp [24] is available at: `https://zenodo.org/records/15025436`

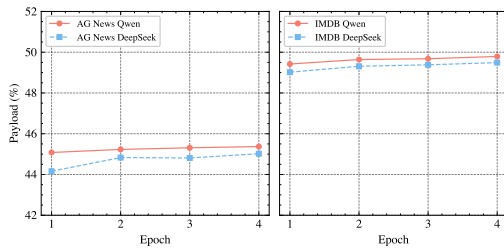
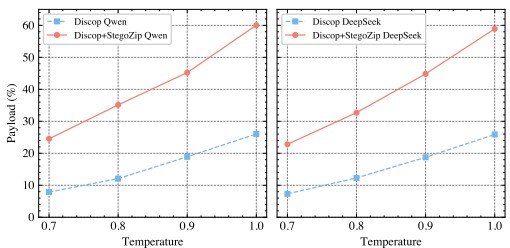

Figure 7: The impact of fine-tuning epochs on the StegoZip payload.

Figure 8: The impact of temperature in stego text generation on StegoZip payload.

# D More Experiment Results

## D.1 Epochs of Fine-tuning

We analyze the impact of fine-tuning epochs for $\mathcal{R}$ to understand the core restoration task, as shown in Figure 7. The results of the AGNews [37] and IMDB [36] (Qwen2.5-7B [32]/DeepSeek-R1-Distill-Llama-8B [33] models) show that the performance saturates after two epochs, with minimal gains beyond, making two epochs optimal for efficiency and effectiveness. Moreover, this implies that the task is relatively simple for the model to learn.

## D.2 Temperature in Generating Stego Texts

We analyze the impact of temperature in stego text generation on the payload performance of StegoZip, as illustrated in Figure 8. Lower temperatures (e.g., 0.7) produce more deterministic outputs, which results in a reduced payload. Conversely, higher temperatures increase the variability in the generated text, thereby enhancing the embedding flexibility and improving the steganographic embedding payload capacity. Notably, the integration of StegoZip with Discop significantly boosts payload performance. Specifically, StegoZip in conjunction with Discop [23] achieves more than double the payload compared with using Discop alone, highlighting the remarkable effectiveness of StegoZip.

## D.3 Visual Quality

**Secret Messages:**

*Treasuries Benefit on Spike in Crude Oil NEW YORK (Reuters) - Treasury debt prices climbed on Monday as investors bet oil prices near record highs might dent U.S. consumption and force the Federal Reserve to slow the pace of interest rate hikes.*

**Stego Text generated by Discop:**

***Ernest William Hornung ( 7 June 1866 – 22 March 1921 ) was an English author and poet known for writing the A. J. Raffles series of stories about a gentleman thief in late 19th @-@ century London.*** *Hornung 's portrait of the city of London and its inhabitants is one that has since come to be recognised as a classic, similar to Dicken's portrayal of the same area. Other reviewers of the same book describe a delicate, yet entertaining character which only Hornung could have portrayed. Chapter 7 – from Raffles of the Palace Guard by Harry H. Burns, who wrote some historical novels of the Scramble for Africa in the 1800s. The return of the Strand and other places of London to Raffles reveals the complex social mix of the city. By the 1890s, London had been transformed into a city for conspicuous consumption, with the rich lording it over the poor. Buildings - as Chancery Lane shows - where mostly taken over by the middle and upper-class and further alienated the working poor, while a few areas, like Covent Garden, remained as the traditional working-class area, with its bars, theatres, and clubs. In terms of the embellishment of London, however, Raffles cannot compete with the grand flamboyance of the Victorians. The atmosphere is still dark though, yet reminds us that it is only dark because Raffles is a figure of light. Raffles himself is described as the main characters of London, a character who thinks for others, and lives by the block like the other hundreds of bachelors, who all have names and individual personalities, except that they do not have the talent to be heroes.Stanley Richards wrote about some of this in his review of Raffles of the Palace Guard : "As*

**Stego Text generated by Discop+StegoZip:**

***Ernest William Hornung ( 7 June 1866 – 22 March 1921 ) was an English author and poet known for writing the A. J. Raffles series of stories about a gentleman thief in late 19th @-@ century London.*** *And a much superior man of letters to most of the other popular novelists who now draw their living from the public .Ïornung published nearly sixty novels and novellas and several short story collections, as well as contributing to some periodicals.His works were internationally popular, and his novels were translated into many languages, including Spanish , French and German .While the origins of the works of Arthur Conan Doyle 's detective Sherlock Holmes have been the subject of some controversy, the background to Hornung 's creation of Raffles has,*

