# OpenReview forum: "StegoZip: Enhancing Linguistic Steganography Payload in Practice with Large Language Models"
_NeurIPS.cc/2025/Conference — NeurIPS 2025 poster_

### Official Review · Reviewer_ykFF · 2025-06-11

**Clarity:** 3
**Significance:** 2
**Originality:** 2
**Rating:** 4
**Confidence:** 3

**Summary:**

This paper improves the payload efficiency of linguistic steganography by preprocessing the secret message using LLMs before embedding. Unlike previous studies that focus solely on optimizing embedding techniques under entropy constraints, the proposed framework (StegoZip) introduces Dynamic Semantic Redundancy Pruning for removing semantically redundant content based on a learned self-information measure, and Index-Based Compression Coding for transforming the pruned message into compact binary codes by token rank-based Huffman coding. Experiments show that StegoZip achieves up to 2.5x higher payload efficiency while maintaining semantic fidelity and steganographic security.

**Questions:**

See Weaknesses.

**Ethical Concerns:**

["NO or VERY MINOR ethics concerns only"]

**Final Justification:**

Since the authors addressed most of our concerns, I decided to raise the rating to 4. However, the authors did not demonstrate that StegoZip does not introduce detectable AIGC-specific artifacts. I hope the authors will consider this in future versions.

**Limitations:**

Yes.

**Paper Formatting Concerns:**

No.

**Quality:**

3

**Strengths And Weaknesses:**

Strengths:

* Leveraging LLMs for semantic pruning and compression is intuitive and makes sense, since LLMs are good at identifying redundant parts and preserving important meanings in a message.
* A formal argument is provided to prove that the security guarantees of the underlying steganographic methods are preserved after integrating StegoZip.
* Extensive evaluations on the commonly used datasets with two SOTA steganographic methods show consistent gains in payload and processing efficiency of StegoZip.

Weaknesses:

* Although the paper evaluates steganalysis performance using three models, these models largely rely on feature extraction or simple architectures. They are insufficient to capture the subtle, high-level distributional patterns potentially introduced by LLM-based generation. To convincingly demonstrate that StegoZip does not introduce detectable AIGC-specific artifacts, the authors should consider training more powerful neural steganalysis models, or simply use GPT as the detector.
* The proposed framework includes a self-checking mechanism where parts of the message that cannot be reliably reconstructed are retained in square brackets. However, the semantic or compression cost introduced by this fallback is not clearly quantified. For example, under default settings, what percentage of content typically requires fallback? How does this affect the effective payload and restoration complexity?
* DSRP claims to preserve key entities through entity detection by assigning them infinite self-information. However, the paper does not clarify how this detection is implemented, nor does it provide any evaluation of its accuracy. If critical entities are missed, important content could be pruned inadvertently, affecting message fidelity.

---

> ### Author Rebuttal · Authors · 2025-07-29
>
> # Responses to Reviewer ykFF
>
> We would like to express our heartfelt gratitude for the time and effort you dedicated to reviewing our paper. Your insightful remarks have prompted us to thoroughly re-examine every aspect of our work. In response, we have carefully prepared the following point-by-point replies, which will be incorporated into the final version:
>
> ### Q1: Although the paper evaluates steganalysis performance using three models, these models largely rely on feature extraction or simple architectures. They are insufficient to capture the subtle, high-level distributional patterns potentially introduced by LLM-based generation. To convincingly demonstrate that StegoZip does not introduce detectable AIGC-specific artifacts, the authors should consider training more powerful neural steganalysis models, or simply use GPT as the detector.
>
> **Response 1:** By leveraging the pseudo-randomization method based on CHACHA20, our method seamlessly integrates with provable secure steganography, inheriting its security guarantees. We have also demonstrated this theoretically (see **Appendix A.2, "StegoZip Maintains the Security of the Underlying Steganographic Algorithm"**). It is important to note that the provable security here refers to being indistinguishable from normal generated text, not from human-written text. Provably secure steganography involves disguising steganographic behavior as normal generated text, as detailed in **Appendix A.1, “Provably Secure Steganography”, and in previous paper [1-5]**.
>
> Thus, for a detector, the challenge lies in distinguishing whether the text is normal generated cover text or stego text generated by embedding secret messages. To evaluate this, we followed prior studies [4-5] and selected three specialized text steganographic analyzer. The classification accuracies of these analyzers were all close to 50%, indicating that the stego text generated under the StegoZip framework is difficult to distinguish, rendering the classifiers equivalent to random guessing.
>
> [1] Ziegler, Zachary, et al. "Neural Linguistic Steganography." *Proceedings of the 2019 Conference on Empirical Methods in Natural Language Processing and the 9th International Joint Conference on Natural Language Processing (EMNLP-IJCNLP)*. 2019.
>
> [2] Yang, Kuan, et al. "Provably secure generative steganography based on autoregressive model." *International Workshop on Digital Watermarking*. Cham: Springer International Publishing, 2018.
>
> [3] Zhang, Siyu, et al. "Provably Secure Generative Linguistic Steganography." *Findings of the Association for Computational Linguistics: ACL-IJCNLP 2021*. 2021.
>
> [3] Ding, Jinyang, et al. "Discop: Provably secure steganography in practice based on" distribution copies"." *2023 IEEE Symposium on Security and Privacy (SP)*. IEEE, 2023.
>
> [4] Wang, Yaofei, et al. "SparSamp: Efficient Provably Secure Steganography Based on Sparse Sampling." *34th USENIX Security Symposium (USENIX Security 25)*. 2025.
>
> ### Q2: The proposed framework includes a self-checking mechanism where parts of the message that cannot be reliably reconstructed are retained in square brackets. However, the semantic or compression cost introduced by this fallback is not clearly quantified. For example, under default settings, what percentage of content typically requires fallback? How does this affect the effective payload and restoration complexity?
>
> **Response 2:** In **Section 4.2: Main Performance of StegoZip, Figure 4** provides a quantitative analysis of the time cost introduced by the self-checking mechanism (i.e., **restoration complexity**). The self-checking process increases StegoZip’s processing time by approximately 15%.
>
> To complement this, we conducted additional experiments under the same settings using the previously fine-tuned Qwen2.5-7B as the restorer, without employing the self-checking mechanism, and 40% of the semantic information was pruned. The results are summarized as follows:
>
> | Dataset | Payload | Rouge_1 (R / C) | BERTScore (R / C) | Fallback Ratio | Missing Ratio |
> | --- | --- | --- | --- | --- | --- |
> | AGNews w/o self-checking | 53.73 | 0.8821 / 0.7833 | 0.9681 / 0.9421 | \ | 0.3239 |
> | AGNews + self-checking | 45.23 | **1.0000** / 0.8485 | **1.0000** / 0.9633 | 0.4945 | 0.1637 |
> | IMDb w/o self-checking | 57.61 | 0.9077 / 0.8465 | 0.9697 / 0.9535 | \ | 0.3634 |
> | IMDb + self-checking | 49.64 | **1.0000** / 0.9445 | **1.0000** / 0.9835 | 0.7238 | 0.1044 |
>
> Here, R / C represents the semantic retention of the restored/compressed secret messages relative to the original. **Fallback Ratio indicates the percentage of content requiring fallback**, while Missing Ratio represents the percentage of missing content per sample.
>
> The results show that while **not using the self-checking mechanism improves the effective payload by 8%, the receiver experiences semantic loss, which is unacceptable in covert communication scenarios.** Thus, a trade-off between the effective payload and restoration complexity is necessary.
>
> With the self-checking mechanism enabled, the AGNews dataset requires recording approximately 50% of missing words on average, while the IMDb dataset requires recording around 72% when 40% of the semantic information is pruned. For context, the average number of words per sample in the AGNews dataset (short text) is 37.63, while in the IMDb dataset (long text) it is 223.56. Thus, even when 16% of words are missing in short texts or 10% in long texts, lossless reconstruction can still be achieved.
>
> ### Q3: DSRP claims to preserve key entities through entity detection by assigning them infinite self-information. However, the paper does not clarify how this detection is implemented, nor does it provide any evaluation of its accuracy. If critical entities are missed, important content could be pruned inadvertently, affecting message fidelity.
>
> **Response 3:** As detailed in **Section 3.3 (Dynamic Semantic Redundancy Pruning)**, to ensure that critical information—such as names, numbers, or other key entities—is preserved and does not compromise the effectiveness of the restoration process, entity detection is performed prior to pruning. This involves a relatively straightforward Named Entity Recognition (NER) task, requiring the training of an entity detector on annotated corpus data.
>
> In our experiments, we utilized the built-in NER pipeline provided by the python-spacy library to detect and tag entities (**approximately 84% accuracy**). The additional time required for this step is minimal (<1%) and is included in the DSRP time shown in Figure 4.
>
> Importantly, even if errors occur during entity detection, the subsequent self-checking mechanism ensures the integrity of the secret message. As we discussed in Question-2 above, when pruning removes certain information, the secret can still be fully reconstructed at the receiving end without any loss, achieving **100% Rouge and BERTScore metrics**. This is demonstrated in Table 2 of Section 4.2: Main Performance of StegoZip.
>
> Thank you once again for your invaluable contributions to this paper. We hope our responses have fully addressed your concerns, and we would be happy to continue the discussion should you have any remaining questions. Wishing you success and fulfillment in both your research and personal endeavors!

---

> > ### Comment · Reviewer_ykFF · 2025-08-04
> >
> > Since the authors addressed most of our concerns, I decided to raise the rating to 4. However, the authors did not demonstrate that StegoZip does not introduce detectable AIGC-specific artifacts. I hope the authors will consider this in future versions.

---

> > > ### Author Response · Authors · 2025-08-04
> > > **Official Response to Reviewer ykFF**
> > >
> > > Thank you so much for taking the time to review our work and for your kind recognition, which truly serves as a great source of encouragement!
> > >
> > > Regarding your concerns about the detectable AIGC-specific artifacts, we will provide further elaboration and clarification in the final version.

---

### Official Review · Reviewer_UhwE · 2025-06-27

**Clarity:** 3
**Significance:** 3
**Originality:** 4
**Rating:** 5
**Confidence:** 4

**Summary:**

This paper introduces a new generative linguistic steganography framework with larger payload than previous methods and faster encoding&decoding speed. In detail, StegoZip consists of two LLM-based components: semantic redundancy pruning and index-based compression coding to achieve the advancing performance. Extensive experiments demonstrate that StegoZip can hide more secret information with faster hiding and decoding efficiency.

**Questions:**

See Strengths And Weaknesses

**Ethical Concerns:**

["NO or VERY MINOR ethics concerns only"]

**Final Justification:**

The authors provide additional results, which help address my concerns, I would keep my score of acceptance and increase my confidence

**Limitations:**

yes

**Quality:**

3

**Strengths And Weaknesses:**

## Strengths
1. The framework is well-engineered, two components are reasonable and effective w.r.t the results.
2. The paper is well-written and the figures and tables are clear

## Weaknesses
1. The paper only considers two linguistic steganography techniques, what about more traditional methods like the one using Huffman/Arithmetic encoding in previous methods [1][2]?
2. I'm wondering if the LLM's capability would impact the final performance --- i.e., leveraging larger or smaller LLMs under the same setting.

[1] Z. -L. Yang, X. -Q. Guo, Z. -M. Chen, Y. -F. Huang and Y. -J. Zhang, "RNN-Stega: Linguistic Steganography Based on Recurrent Neural Networks," in IEEE Transactions on Information Forensics and Security, vol. 14, no. 5, pp. 1280-1295, May 2019, doi: 10.1109/TIFS.2018.2871746.

[2] Ziegler Z M, Deng Y, Rush A M. Neural linguistic steganography[J]. arXiv preprint arXiv:1909.01496, 2019.

---

> ### Author Rebuttal · Authors · 2025-07-29
>
> # Responses to Reviewer UhwE
>
> We sincerely appreciate the time and effort you spent to reviewing our paper. Your insightful feedback has motivated us to thoroughly re-evaluate every aspect of our work. Below, we have provided detailed responses and will incorporate these revisions into the final version of the manuscript:
>
> ### Q1: The paper only considers two linguistic steganography techniques, what about more traditional methods like the one using Huffman/Arithmetic encoding in previous methods?
> **Response 1:** StegoZip is designed as a plug-and-play secret messages compressor that can be integrated with any downstream linguistic steganography scheme as long as that scheme accepts a raw bitstream and returns a stego text. We tried combining StegoZip with two representative arithmetic-coding–based methods: AC [1] and Meteor [2]. The experimental results are as follows:
>
> | Steganography | Payload (%) ↑  | Encoding Time (s) ↓ | Decoding Time (s) ↓ |
> | --- | --- | --- | --- |
> | AC [1] | 20.02 | 17.08 | 17.25 |
> | AC [1] + StegoZip | 45.01 (↑24.99) | 13.17 (↓3.91) | 13.27 (↓3.98) |
> | Meteor [2] | 11.27 | 36.20 | 36.58 |
> | Meteor [2] + StegoZip | 31.49 (↑20.22) | 22.46 (↓13.74) | 22.08 (↓14.50) |
>
> Due to the rebuttal schedule, we only randomly sampled 100 news items from AGNews to serve as the secret messages and, for stego prompts, extracted the first two sentences of 100 randomly chosen samples from WikiText dataset. We use the previously fine-tuned Qwen2.5-7B as the restorer while all other settings are the same as those in the Baselines. When integrated with AC-based steganographic methods under lossless decoding, StegoZip still achieves 2× the payload of the baselines while requiring less processing time in practice.
>
> [1] Ziegler, Zachary, et al. "Neural Linguistic Steganography." *Proceedings of the 2019 Conference on Empirical Methods in Natural Language Processing and the 9th International Joint Conference on Natural Language Processing (EMNLP-IJCNLP)*. 2019.
>
> [2] Kaptchuk, Gabriel, et al. "Meteor: Cryptographically secure steganography for realistic distributions." *Proceedings of the 2021 ACM SIGSAC Conference on Computer and Communications Security*. 2021.
>
> ### Q2: I'm wondering if the LLM's capability would impact the final performance --- i.e., leveraging larger or smaller LLMs under the same setting.
>
> **Response 2:** We have analyzed this problem in Table 3 ("Efficiency comparison of LLMs of varying sizes in compression and restoration"). As the model size increases, the achievable payload consistently rises; however, both encoding and decoding times increase with model size.
>
> Thank you once again for your invaluable contributions to this paper. We hope our responses address your concerns fully, and we would be delighted to engage in further discussions should you have any additional questions. Wishing you continued success in both your research and life!

---

> > ### Comment · Reviewer_UhwE · 2025-07-31
> >
> > Thank you for your additional results, I would keep my score of acceptance.

---

> > > ### Author Response · Authors · 2025-08-01
> > > **Official Response to Reviewer UhwE**
> > >
> > > Thank you for reviewing our work and your recognition—it’s a great encouragement to us!

---

### Official Review · Reviewer_jVW7 · 2025-06-28

**Clarity:** 3
**Significance:** 2
**Originality:** 2
**Rating:** 4
**Confidence:** 2

**Summary:**

The paper introduces StegoZip, a framework designed to enhance the payload capacity of linguistic steganography using large language models (LLMs). StegoZip addresses the challenge of low secret payload in generative steganography by optimizing secret message processing through two main components: Dynamic Semantic Redundancy Pruning (DSRP) and Index-Based Compression Coding (ICC). DSRP leverages LLMs to remove low-information elements from secret messages, while ICC compresses the pruned content into compact binary codes. The framework integrates with state-of-the-art (SOTA) steganographic methods, achieving a 2.5x increase in payload compared to baselines while maintaining comparable processing times and ensuring lossless message reconstruction. The paper evaluates StegoZip using public datasets (IMDb, AGNews, WikiText-2) and LLMs like Qwen2.5-7B and DeepSeek-R1-DistillLama-8B, demonstrating improvements in efficiency and semantic retention. The work also discusses its potential for secure covert communication, with applications in privacy-preserving scenarios.

**Questions:**

- The self-checking process in DSRP (Page 5) is described as ensuring lossless reconstruction by replacing unaligned portions with original content. Could the authors provide a more detailed explanation or example of this process, including its computational cost?
- Robustness to ambiguous tokenization. Please re-run payload/accuracy on sentences that do contain BPE ambiguity or report a mitigation cost when you fall back to character-level tokenizers. A large degradation would cast doubt on practical use.

**Ethical Concerns:**

["NO or VERY MINOR ethics concerns only"]

**Final Justification:**

The authors have addressed my concerns.

**Limitations:**

Yes,

**Paper Formatting Concerns:**

Minor typos: “epoches” in Appendix D.

**Quality:**

2

**Strengths And Weaknesses:**

Strengths:
- StegoZip uses LLMs for semantic pruning and compression, achieving a significant payload increase (2.5x over baselines).
- The technical quality is good, with a clear methodology supported by detailed descriptions of DSRP and ICC, including mathematical formulations (e.g., self-information calculation, $\alpha$-quantile pruning).

Weaknesses:
- The paper claims broad applicability for covert communication but does not deeply explore domain-specific challenges (e.g., handling technical jargon in medical domains or low-resource languages).
- Reported 2.5× gain is measured only on English news and movie reviews; no multilingual, chat-style, or code-mixed corpora are tested, limiting external validity

---

> ### Author Rebuttal · Authors · 2025-07-30
>
> # Responses to Reviewer jVW7
>
> We sincerely appreciate the time and effort you invested in reviewing our paper. Your insightful comments have prompted us to conduct a comprehensive re-evaluation of every aspect of our work. Below, we have carefully prepared point-by-point responses, which will be incorporated into the final version:
>
> ### Q1: The paper claims broad applicability for covert communication but does not deeply explore domain-specific challenges (e.g., handling technical jargon in medical domains or low-resource languages).
> **Response 1:** We would like to clarify that the applicability of StegoZip, as discussed in the paper, specifically refers to its adaptability to integrable text steganography algorithms. We hope this explanation helps to resolve any misunderstanding.
>
> Moreover, we acknowledge the importance of the domain-specific challenges you mentioned on covert communication. To address this, we have added test results on the MedNLI dataset from the medical domain [1]. Specifically, we randomly selected 1,000 test samples and used the previously fine-tuned Qwen2.5-7B model on the English AGNews dataset as the Restorer (i.e., this test evaluates out-of-distribution data) and Discop as the underlying steganographic algorithm. All other settings remain consistent with the baseline descriptions. The experimental results are as follows:
>
> | Dataset & Steganography | Payload (%) ↑ | Encoding Time (s) ↓ | Decoding Time (s) ↓ |
> | --- | --- | --- | --- |
> | MedNLI | 21.51 | 15.21 | 15.07 |
> | MedNLI + Stegozip | 33.79 (↑8.28) | 11.71 (↓3.5) | 11.76 (3.31) |
>
> Even when dealing with out-of-distribution data that includes a lots of technical jargon, StegoZip can still achieve an 8% increase in payload while reducing processing time without re-fine-tuning. Initially, fine-tuning the small-scale LLMs was intended solely to enable them to comprehend the restoration task, without considering the specific domain of the secret messages. However, given the specialized nature of covert communications, interacting parties can often anticipate the domain of the secret messages and apply fine-tuning, as discussed in Section 4.4, “Cross-Domain Transferability Analysis”.
>
> [1] Romanov, Alexey, and Chaitanya Shivade. "Lessons from natural language inference in the clinical domain." arXiv preprint arXiv:1808.06752 (2018).
>
> ### Q2: Reported 2.5× gain is measured only on English news and movie reviews; no multilingual, chat-style, or code-mixed corpora are tested, limiting external validity.
>
> **Response 2:** We selected English news and movie reviews as test data because they encompass a broad range of knowledge, making them more representative as secret messages in covert communication scenarios. However, we greatly value the domain-specific challenges you raised and have taken the initiative to explore more challenging tasks.
>
> To this end, we used the previously fine-tuned Qwen2.5-7B model on the English AGNews dataset as the Restorer and Discop as the underlying steganographic algorithm. For the experiments, we randomly selected 1,000 French dialogue samples from the Claire French Dialogue Dataset (CFDD) [2], using the first two dialogues of each sample as the secret messages. Additionally, we constructed a Code-Mixed dataset by randomly selecting 1,000 code statements across MySQL, Python, C++, and Java. **All test data represent out-of-distribution samples.** Other experimental settings were kept consistent with the baseline descriptions. The experimental results are as follows:
>
> | Dataset & Steganography | Payload (%) ↑  | Encoding Time (s) ↓ | Decoding Time (s) ↓ |
> | --- | --- | --- | --- |
> | CFDD | 18.53 | 46.16 | 46.13 |
> | CFDD + Stegozip | 39.90 (↑21.37) | 34.57 (↓11.59) | 33.17 (↓12.96) |
> | Code | 20.44 | 12.95 | 13.11 |
> | Code + Stegozip | 23.64 (↑3.20) | 11.68 (↓1.27) | 11.00 (↓2.11) |
>
> Even with a Restorer fine-tuned on an English dataset, the system performed exceptionally well on the French test set, achieving **lossless decoding in less time** and more than **doubling the payload**. This aligns with our design goal of fine-tuning the Restorer to understand the restoration task itself, rather than data from a specific domain.
>
> However, the performance on the Code-Mixed dataset was less impressive. This is primarily due to the nature of the dataset: code statements are relatively short, with many resembling examples such as `float updateInfo(int val_qliz) { return 84.53f; }`. The average word count in the dataset is only around 15, which limits its compression capacity. It is worth noting that **StegoZip is primarily designed to handle large volumes of secret messages**, such as reports, logs, or multimedia metadata. These types of messages, if left unprocessed, would typically require multiple transmissions or result in suspiciously long stego text. This aligns with common application scenarios in covert communication, where handling substantial payloads is a critical requirement.
>
> [2] Hunter, Julie, et al. "The claire french dialogue dataset." *arXiv preprint arXiv:2311.16840* (2023).
>
> ### Q3: The self-checking process in DSRP (Page 5) is described as ensuring lossless reconstruction by replacing unaligned portions with original content. Could the authors provide a more detailed explanation or example of this process, including its computational cost?
>
> **Response 3:** **In Figure 2 of the paper, we illustrate a specific example of this process.** The original secret message, denoted as $m$, undergoes processing through DSRP and is transformed into $m_c$. At this stage, a self-checking mechanism is applied, producing $m_r$. If errors are detected during the reconstruction of the second and third missing words, the corrected words are enclosed in brackets [ ], resulting in $m_c’$. The receiver obtains $m_c’$ without any loss of information and can clearly identify the words enclosed in brackets. This allows $m_c’$ to be split into $m_c$ and an error correction table, which is then used to resolve the errors in the second and third missing words.
>
> Since the restorer and $m_c$ are aligned, the receiver will encounter the same reconstruction errors in the second and third words ($m_r$) when attempting to recreate the original secret message. These errors can subsequently be corrected using the error correction table, ultimately allowing the receiver to fully reconstruct the original secret message, $m$. Specific examples are as follows:
>
> - $m$: Toyota to open south China plant Japan carmaker Toyota enters a joint venture to produce saloon cars in southern China.
> - $m_c$: Toyota [] [] [] China plant Japan carmaker Toyota enters a [] [] to produce saloon cars [] southern China.
> - $m_r$: Toyota [to] **[establish] [a]** China plant Japan carmaker Toyota enters a [joint][venture] to produce saloon cars [in] southern China.
> - $m_c’$: Toyota [] **[open] [south]** China plant Japan carmaker Toyota enters a [] [] to produce saloon cars [] southern China.
>
> In **Section 4.2: Main Performance of StegoZip**, Figure 4 provides a quantitative analysis of the time cost introduced by the self-checking mechanism. The self-checking process increases StegoZip’s processing time by approximately 15%.
>
> ### Q4: Robustness to ambiguous tokenization. Please re-run payload/accuracy on sentences that do contain BPE ambiguity or report a mitigation cost when you fall back to character-level tokenizers. A large degradation would cast doubt on practical use.
>
> **Response 4:** Regarding the robustness issue caused by tokenization ambiguity, we have discussed this in **Appendix C.5: "Token Ambiguity"**. First, it is important to note that current mainstream LLMs (e.g., GPT, LLaMA, Qwen, etc.) predominantly use BPE-based tokenizers rather than character-level tokenizers, which are prone to significant semantic loss. However, the vocabulary of BPE-based tokenizers is not prefix-free, meaning that the same input sequence may have multiple tokenization paths.
>
> It is worth clarifying that this **ambiguity primarily affects the underlying steganographic algorithms integrated into the StegoZip framework, rather than StegoZip itself.** To mitigate information loss caused by such ambiguity, existing steganographic systems commonly employ a "disambiguation" strategy [3]. To evaluate the robustness of StegoZip, we tested the disambiguating steganographic algorithm SyncPool [3] under the same experimental settings in the paper. The results are as follow:
>
> | Steganography | Payload (%) ↑  | Encoding Time (s) ↓ | Decoding Time (s) ↓ |
> | --- | --- | --- | --- |
> | SyncPool | 8.27 | 29.31 | 28.77 |
> | SyncPool + StegzoZip | 17.22 (↑8.95) | 23.09 (↓6.22) | 23.51 (↓5.26) |
>
> Although there is a slight decline in performance for both the baseline and our approach, the steganographic algorithm integrated into the StegoZip framework still achieves a 2× increase in payload while requiring less processing time compared to the baseline method.
>
> [3] Qi, Yuang, et al. "Provably secure disambiguating neural linguistic steganography." *IEEE Transactions on Dependable and Secure Computing* (2024).
>
> Thank you once again for your valuable contributions to this paper. We hope our responses have addressed your concerns and would be happy to continue the discussion should you have any remaining questions. Wishing you success and prosperity in both your research and personal endeavors.

---

> ### Author Response · Authors · 2025-08-05
> **Response to Reviewer jVW7**
>
> Thank you so much for  your feedback. We’d like to know if you have any further concerns or suggestions regarding the revised content.

---

### Official Review · Reviewer_ovAh · 2025-07-03

**Clarity:** 2
**Significance:** 2
**Originality:** 2
**Rating:** 3
**Confidence:** 3

**Summary:**

This paper presents a framework to first compress secret text and then passes the compressed text for linguistic steganographic systems, which achieves better payload. Experiments on two datasets show the effectiveness of StegoZip.

**Questions:**

1. What is the core technical contribution of this work? It seems to me the steganography algorithm is based on ChaCha20 so the main contribution is a method to compress the secret message?

2. How StegoZip generalizes to secret message with very little or no semantic redundancy? Considering the common application scenario of using steganography, it very likely the secret message is short and contains only key information like "XXX person at YYY time/location will do ZZZ", if you keep all the key entities, there are not much thing left to compress.

3. Why we need to train the restorer? Can we simply just use a standard out-of-the-shelf LLM?

4. What is the "prefix (as prior)" in Figure 2 Step 2: ICC (and also Step 4 Message Decoding).

5. Have you tried combining StegoZip with those arithmetic coding based stegonagraphic algorithms?

**Ethical Concerns:**

["NO or VERY MINOR ethics concerns only"]

**Final Justification:**

I have read the author's rebuttal and decided to keep my original score. My main concern is the limited technical contributions and application scope of proposed method.

**Limitations:**

yes

**Quality:**

2

**Strengths And Weaknesses:**

Strengths:
1. Study a practical and interesting problem.
2. The proposed framework is overall reasonable.

Weaknesses:
1. The novelty and core contributions of this framework are limited.
2. Some paper presentations can be improved.

---

> ### Author Rebuttal · Authors · 2025-07-29
>
> # Responses to Reviewer ovAh
> We sincerely appreciate your time to reviewing our paper. Your insightful comments have prompted us to thoroughly re-examine every aspect of our work, and we have carefully prepared the following point-by-point responses and will make these changes in the final version:
>
> ### Q1: What is the core technical contribution of this work? It seems to me the steganography algorithm is based on ChaCha20 so the main contribution is a method to compress the secret message?
>
> **Response 1:** The core technical contribution of this paper lies in leveraging the advanced comprehension and predictive capabilities of LLMs to design a secret messages compression method for covert communication. Chacha20 serves solely as the pseudo-randomization processing method applied to the compressed messages, with the purpose of enabling compatibility with mainstream provably secure steganography methods. Through the proposed Stegozip framework, the payload of a single transmission is significantly increased (about 2.5× the payload of the baseline methods), effectively mitigating the behavioral abnormalities of frequent transmissions in traditional systems caused by their low payload, while ensuring information integrity.
>
> ### Q2: How StegoZip generalizes to secret message with very little or no semantic redundancy? Considering the common application scenario of using steganography, it very likely the secret message is short and contains only key information like "XXX person at YYY time/location will do ZZZ", if you keep all the key entities, there are not much thing left to compress.
>
> **Response 2:** On the one hand, our ablation study in Section 4.5 (Table 5, w/o DSRP) demonstrates that StegoZip still achieves a ≈10% increase in payload, even without removing semantic redundancy. This is because the LLM’s predictive capability alone can re-encode the literal content into a more compact bit-stream. On the other hand, the primary design objective of StegoZip is to handle bulk secrets—such as reports, logs, or multimedia metadata—whose size would otherwise necessitate multiple transmissions or suspiciously long stego texts. This is a common application scenario in covert communication. Our information compression ratio is considerable, as demonstrated by achieving 2.5× the payload of the baseline methods while maintaining comparable processing times in practice.
>
> ### Q3: Why we need to train the restorer? Can we simply just use a standard out-of-the-shelf LLM?
>
> **Response 3:** Given the specific requirements of covert communication scenarios, using large cloud-based models is not feasible, as we cannot entrust the processing or restoration of secret messages to the cloud. In practical covert communication environments, the hardware’s computing power is often limited, such as on laptops or mobile devices. Under these constraints, we can only utilize models with smaller parameter sizes, such as no larger than 7B models. However, standard models of this size struggle to effectively understand the task of secret message restoration. To address this, we fine-tune a basic LLM, transforming it into a lightweight and privately deployable solution for restoring secret messages. More discussions are shown in Section 3.2 "Private Restorer in StegoZip Framework".
>
> ### Q4: What is the "prefix (as prior)" in Figure 2 Step 2: ICC (and also Step 4 Message Decoding).
>
> **Response 4:** The “prefix” refers to the fixed system instruction used to fine-tune the restorer model (see Figure 3). This identical instruction is prepended to every secret message before it enters the ICC encoder. By doing so, it provides prior knowledge that the subsequent text represents a low-semantic structure, enabling the model to compress it further into compact binary codes. In Section 4.5 (Table 5, w/o instruction in ICC), we ablate this design. When the prefix was omitted, the payload decreased by approximately 3%, highlighting its necessity.
>
> ### Q5: Have you tried combining StegoZip with those arithmetic coding based stegonagraphic algorithms?
>
> **Response 5:** Yes. StegoZip is designed as a plug-and-play secret messages compressor that can be integrated with any downstream linguistic steganography scheme as long as that scheme accepts a raw bitstream and returns a stego text. We tried combining StegoZip with two representative arithmetic-coding–based methods: AC [1] and Meteor [2]. The experimental results are as follows:
>
> | Steganography | Payload (%) ↑  | Encoding Time (s) ↓ | Decoding Time (s) ↓ |
> | --- | --- | --- | --- |
> | AC [1] | 20.02 | 17.08 | 17.25 |
> | AC [1] + StegoZip | 45.01 (↑24.99) | 13.17 (↓3.91) | 13.27 (↓3.98) |
> | Meteor [2] | 11.27 | 36.20 | 36.58 |
> | Meteor [2] + StegoZip | 31.49 (↑20.22) | 22.46 (↓13.74) | 22.08 (↓14.50) |
>
> Due to the rebuttal schedule, we only randomly sampled 100 news items from AGNews to serve as the secret messages and, for stego prompts, extracted the first two sentences of 100 randomly chosen samples from WikiText dataset. We use the previously fine-tuned Qwen2.5-7B as the restorer while all other settings are the same as those in the Baselines. When integrated with AC-based steganographic methods under lossless decoding, StegoZip still achieves 2× the payload of the baselines while requiring less processing time in practice.
>
> [1] Ziegler, Zachary, et al. "Neural Linguistic Steganography." *Proceedings of the 2019 Conference on Empirical Methods in Natural Language Processing and the 9th International Joint Conference on Natural Language Processing (EMNLP-IJCNLP)*. 2019.
>
> [2] Kaptchuk, Gabriel, et al. "Meteor: Cryptographically secure steganography for realistic distributions." *Proceedings of the 2021 ACM SIGSAC Conference on Computer and Communications Security*. 2021.
>
> Thank you once again for your invaluable efforts on this paper. We hope that our responses address your concerns, and we would be delighted to continue the discussion if any questions remain. Wishing you all the best in both life and research ^v^.

---

### Note · Authors · 2025-08-12

We sincerely thank the reviewers and the ACs for their constructive feedback and recognition of our work's strengths. In the original submission, the reviewers highlighted several key aspects:

1. **Practicality and Innovation:** The paper addresses a practical problem by leveraging LLMs for semantic pruning, effectively enhancing steganographic payloads. (Reviewer ovAh and jVW7)
2. **Reasonable Framework:** The framework is reasonable, achieving a significant 2.5x payload increase with clear methodology and through consistent experimental improvements. （All Reviewers)
3. **Writing and Presentation:** The paper is clear and well-engineered, with formal proofs, clear visuals, and comprehensive experiments demonstrating its advantages. (Reviewer jVW7 and UhwE)

We have carefully addressed all the concerns raised during the rebuttal:

1. **Clarity of Contributions:** We highlighted the significance of StegoZip that substantially increases the payload of a single transmission (approximately 2.5× higher than baseline methods), mitigating behavioral abnormalities in traditional low-payload systems while ensuring robust information integrity and compatibility with secure steganography methods.
2. **Detailed Explanations:** We clarified the roles of methodology, as well as the distinction between steganalysis and generated text detection.
3. **Expanded Discussion and Comparisons:** We conducted additional experiments to address domain-specific challenges, external validity, BPE ambiguity, and applicability to other steganographic algorithms.

These revisions enhance the paper's clarity and further validate StegoZip's applicability without altering its core contributions or novelty. **We are particularly encouraged by the positive feedback from reviewers UhwE and ykFF, who acknowledged the improvements and provided positive ratings. For the other two reviewers, we carefully addressed their concerns through additional experiments and more detailed explanations during the rebuttal. However, no further detailed feedback was received, possibly due to time constraints or workload. We hope they acknowledge our efforts and the significant improvements made to the manuscript, and we sincerely look forward to a discussion between the ACs and reviewers.**

We believe the revised manuscript offers a rigorous and novel solution to secure and practical linguistic steganography. Thank you for your time and consideration.

---

### Decision · Program_Chairs · 2025-09-17

**Decision:**

Accept (poster)

**Comment:**

The paper introduces StegoZip, a framework designed to enhance the payload capacity of linguistic steganography using large language models (LLMs). StegoZip consists of two main components: Dynamic Semantic Redundancy Pruning (DSRP) and Index-Based Compression Coding (ICC). DSRP remove slow-information elements from secret messages, and ICC compresses the pruned content into compact binary codes. The framework achieves a 2.5x increase in payload compared to baselines.

Strengths:
Strong experiment results (2x payload increase).

Weaknesses:
The experiments can be conducted on more diverse scenarios to further illustrate its broad usability and generality (e.g., domain-specific, multilingual, code-mixed, conversational, etc)